# A thalamocortical circuit for updating action-outcome associations

**Virginie Fresno[1,2], Shauna L Parkes[1,2], Angélique Faugère[1,2], Etienne Coutureau[2†]\*, Mathieu Wolff[2†]\***

[1]CNRS, INCIA, UMR 5287, Bordeaux, France; [2]Université de Bordeaux, INCIA, UMR 5287, Bordeaux, France

**Abstract** The ability to flexibly use knowledge is one cardinal feature of goal-directed behaviors. We recently showed that thalamocortical and corticothalamic pathways connecting the medial prefrontal cortex and the mediodorsal thalamus (MD) contribute to adaptive decision-making (Alcaraz et al., 2018). In this study, we examined the impact of disconnecting the MD from its other main cortical target, the orbitofrontal cortex (OFC) in a task assessing outcome devaluation after initial instrumental training and after reversal of action-outcome contingencies. Crossed MD and OFC lesions did not impair instrumental performance. Using the same approach, we found however that disconnecting the OFC from its other main thalamic afferent, the submedius nucleus, produced a specific impairment in adaptive responding following action-outcome reversal. Altogether, this suggests that multiple thalamocortical circuits may act synergistically to achieve behaviorally relevant functions.
DOI: https://doi.org/10.7554/eLife.46187.001

\*For correspondence:
etienne.coutureau@u-bordeaux.fr
(EC);
mathieu.wolff@u-bordeaux.fr
(MW)

†These authors contributed equally to this work

**Competing interests:** The authors declare that no competing interests exist.

## Introduction

In dynamic environments, the ability to engage in adaptive behaviors is essential to meet basic needs and desires. This sometimes requires updating the current understanding of rules governing events or actions. A large literature has documented that such goal-directed behaviors rely on highly flexible cognitive processes that appear to be largely supported by prefrontal cortical areas (*O'Doherty et al., 2017*; *Parkes and Coutureau, 2018*). However, mounting experimental evidence indicates that functional interactions between prefrontal and subcortical areas are essential to support these functions (*Balleine and O'Doherty, 2010*; *Verschure et al., 2014*). In particular, the contribution of several thalamic nuclei is now better understood and acknowledged in current views on the functioning of thalamocortical circuits (*Pergola et al., 2018*; *Rikhye et al., 2018b*; *Wolff and Vann, 2019*).

Recently, the interactions between the medial prefrontal cortex (mPFC) and the mediodorsal thalamus (MD) have been highlighted in multiple behavioral setups and through various causal interventions (*Bradfield et al., 2013*; *Browning et al., 2015*; *Bolkan et al., 2017*; *Schmitt et al., 2017*; *Alcaraz et al., 2018*; *Marton et al., 2018*; *Rikhye et al., 2018a*). For instance, we recently reported that thalamocortical and corticothalamic pathways connecting the mPFC and the MD support complementary but dissociable aspects of goal-directed behavior (*Alcaraz et al., 2018*). However, the mPFC is not the only cortical recipient of MD projections as the orbitofrontal cortex (OFC) is also one of its main cortical targets (*Groenewegen, 1988*; *Alcaraz et al., 2016*). Interestingly, while the OFC has long been associated with stimulus-outcome predictions (*Balleine et al., 2011*; *McDannald et al., 2014*; *Stalnaker et al., 2014*), its contribution to flexible goal-directed responding has now also been highlighted (*Gremel and Costa, 2013*; *Bradfield et al., 2015*; *Bradfield et al., 2018*; *Panayi and Killcross, 2018*; *Parkes et al., 2018*). Beyond MD afferents, the OFC is also the recipient of a massive projection from the submedius thalamus (Sub) (*Yoshida et al.,*

*1992*; *Alcaraz et al., 2015*; *Kuramoto et al., 2017*). While this region is currently poorly understood, it was recently demonstrated to support the updating of stimulus-outcome associations, in a manner that was highly reminiscent of OFC functions (*Alcaraz et al., 2015*). This suggests that a complex interplay occurs between the OFC and its thalamic afferents when flexible responding is essential.

Thus, in the present study, we aimed to build upon our previous study (*Alcaraz et al., 2018*) and to disentangle the functional interactions of the OFC with its two main thalamic afferents. To this end, we used an instrumental training paradigm allowing to test for initial action-outcome encoding but also its reversal (*Parkes et al., 2018*). The latter forces rats to engage in flexible responding. We then assessed the impact of disconnecting the OFC either from its MD afferent (Experiment 1) or from its Sub afferent (Experiment 2). We found that disconnecting the OFC from the MD left instrumental performance unaltered throughout all phases of testing. Interestingly, disconnecting the OFC from the Sub produced a marked and selective impairment in goal-directed responding following action-outcome reversal. The current set of results, together with prior evidence (*Alcaraz et al., 2018*), indicate that distinct thalamocortical circuits may act in parallel as a function of current behavioral demand.

## Results

### Histology

To disconnect the OFC from its two main thalamic afferents, we performed unilateral lesions of either the OFC and the MD, or the OFC and the Sub. Critically, lesions were made in the same hemisphere (IPSI) or contralateral hemispheres (CONTRA). It is expected that in the CONTRA groups, communication between cortical and thalamic areas should be interrupted while communication remains possible in the IPSI groups. All lesion placements were counterbalanced across hemispheres. To include rats in the behavioral analyses, lesions performed at both the cortical and the thalamic levels needed to be accurate. We used the same criteria for IPSI and CONTRA groups, thus ensuring that the overall cortical and thalamic damage was equivalent in these groups.

The cortical lesions were highly similar to our previous work (*Alcaraz et al., 2015*). In general, the lesions targeted mostly the lateral (LO) and the ventral (VO) parts of the orbitofrontal cortex and in most cases the medial orbital region (MO) was intact. A representative example is shown in *Figure 1A* and the extent of the lesions given in *Figure 1D*. Some rats were excluded because cortical lesions were too small or too dorsal, leaving a substantial portion of the LO and VO intact.

NMDA injections within the MD produced variable damage (*Figure 1B and E*). Inclusions harbored significant damage encompassing the three segments of the MD and, in many instances, the adjacent intralaminar nuclei (mostly the paracentral and the centrolateral nuclei). Some of the rats however had only minimal thalamic damage because the lesions were too dorsal and were therefore excluded. As the lesions were unilateral, we generally succeeded in preserving midline thalamic nuclei and even the paraventricular nucleus was intact for included rats, which is difficult to achieve with bilateral lesions. The habenula however was damaged in many cases. Sub lesions (*Figure 1C and F*) were largely in line with previous work (*Alcaraz et al., 2015*) and highly specific. The *reuniens*/rhomboïd complex was unaffected in the majority of included rats while a few displayed moderate damage at this level. Importantly, thalamic damage did not overlap between the MD and the Sub for any of the included rats.

After histological examination by two experimenters blind to behavioral data, a total of 15 rats out of 48 were excluded from behavioral analyses (5 OFC-MD-IPSI, 5 OFC-MD-CONTRA, 3 OFC-Sub-IPSI and 2 OFC-Sub-CONTRA). The final group sizes for the OFC-MD groups were: 7 IPSI and 7 CONTRA rats and for the OFC-Sub groups: 9 IPSI and 10 CONTRA rats.

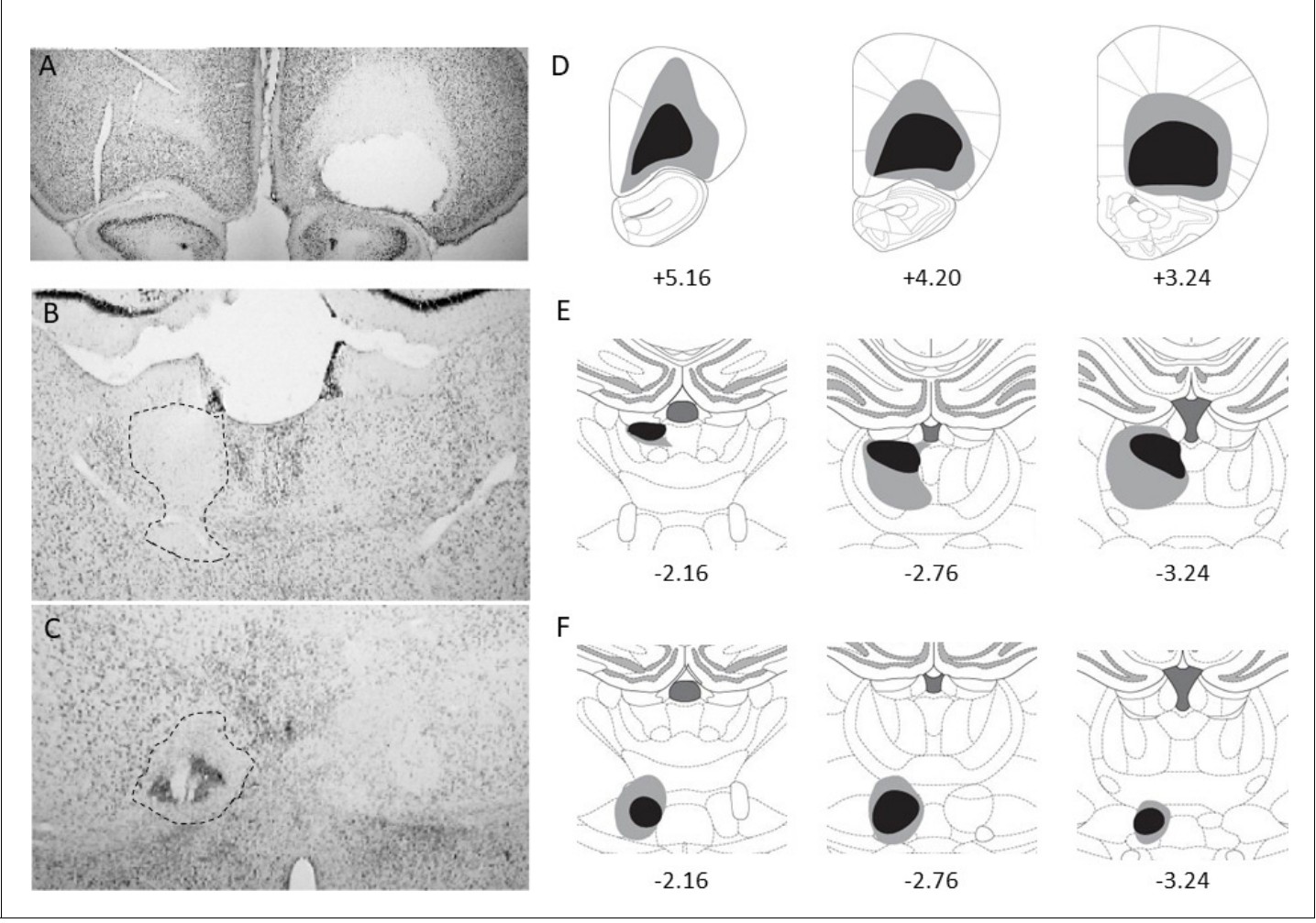

**Figure 1.** Unilateral cortical and thalamic lesions. Photomicrographs of thionin-stained coronal sections showing representative OFC (**A**), MD (**B**) and Sub (**C**) lesions. A dotted line highlights thalamic lesions. The extent of each lesion for the included rats is shown on the right (**D–F**) with the largest lesion shown in gray and the smallest in black (from top to bottom: OFC, MD and Sub). Numbers indicate approximate location relative to bregma (in mm).

DOI: https://doi.org/10.7554/eLife.46187.002

## Experiment 1: Disconnecting the orbitofrontal cortex from the mediodorsal thalamus

A schematic for the behavioral paradigm is illustrated in *Figure 2*, showing the successive phases consisting in instrumental training, outcome devaluation 1, outcome reversal training and outcome devaluation 2. The same paradigm was used in experiments 1 and 2.

### Instrumental training

Instrumental conditioning gradually developed over the 10 consecutive training sessions. As evident in *Figure 3A*, learning appeared to establish at similar rates in IPSI and CONTRA groups even though asymptotic performance appeared somewhat lower in the CONTRA group. The statistical analysis supported these observations as the ANOVA revealed a significant effect of Session ($F_{(9,108)} = 69.5$, $p<0.0001$) but the effect of Lesion only approached significance ($F_{(1,12)} = 3.5$, $p=0.0853$). Importantly, Lesion and Session did not interact ($F_{(9,108)} = 1.0$, $p=0.4215$), suggesting a similar rate of instrumental acquisition in both groups.

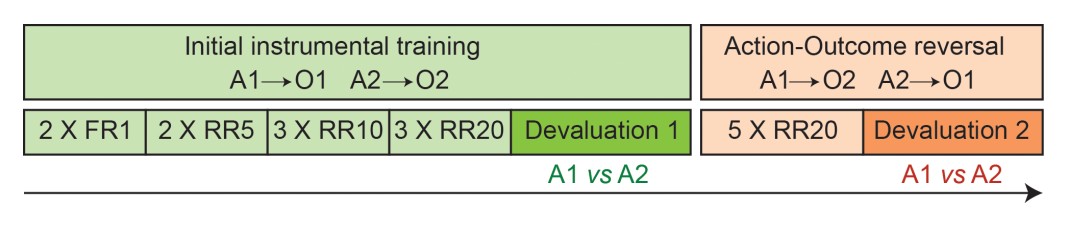

**Figure 2.** Experimental paradigm to assess flexible goal-directed responding. The design is adapted from *Parkes et al. (2018)* and consists in providing a choice test following outcome devaluation after an initial instrumental training phase (green), but also after reversal of action-outcome contingencies (orange).

DOI: https://doi.org/10.7554/eLife.46187.003

## Outcome devaluation

*Figure 3b* shows the test conducted under extinction conditions, immediately after outcome devaluation. During sensory-specific satiety, rats from the IPSI group consumed more food than rats from the CONTRA group (IPSI: 9.9 ± 3.8 g, CONTRA: 5.5 ± 2.6 g, $F_{(1,12)} = 6.5$, p=0.0254). Nevertheless, during the subsequent choice test, both groups expressed a marked preference for the lever associated with the still valued outcome ($F_{(1,12)} = 37.8$, p<0.0001). This preference did not vary as a

**Figure 3.** Disconnecting the OFC and the MD. Mean rate of lever presses (± sem) during instrumental conditioning (A) or during action-outcome contingency reversal (D). Lever presses (% Baseline,+ sem) during the initial choice test (B) or following contingency reversal (E). Consumption tests following the initial choice test (C) and the post-reversal test (F).

DOI: https://doi.org/10.7554/eLife.46187.004

function of lesion status as neither the main effect of Lesion (F(1,12) = 1.9, p=0.1957) nor the Lesion X Devaluation interaction (F(1,12) = 0.0, p=0.9530) approached significance. The post-test consumption assays shown in *Figure 3C* confirmed that devaluation was effective for both groups and that rats in the IPSI and CONTRA groups consumed a similar amount for both the valued and the devalued outcomes (Devaluation, F(1,12) = 16.5, p=0.0016; Lesion, Lesion X Devaluation, F < 1).

## Reversal

After the devaluation test, rats were given five RR20 instrumental training sessions, during which the action-outcome contingencies were reversed. During that phase, all rats maintained a high level of instrumental responding, comparable to that attained at the end of the initial instrumental phase (*Figure 3D*). Responding slightly increased during these five sessions, in a comparable manner across all groups. The ANOVA revealed a significant effect of Session (F(14, 48)=8.1, p<0.0001) but not of Lesion (F < 1) or Lesion X Session interaction (F(4,68) = 1.1, p=0.3781). Thus, during reversal training, there was no evidence that instrumental responding differed between groups.

During outcome devaluation, all rats consumed similar amounts of food (IPSI: 11.0 ± 2.4 g, CONTRA: 9.2 ± 3.0 g, F(1,12) = 1.6, p=0.2361). During the choice test conducted under extinction conditions (*Figure 3E*), all rats expressed a marked preference for the action leading to the still valued outcome, showing that they succeeded in updating action-outcome contingencies. These observations were supported by statistical analyses. We found a main effect of Devaluation (F(1,12) = 11.9, p=0.0048) but no effect or Lesion or Lesion X Devaluation interaction (Fs <1). Similar conclusions arose from the analysis of post-test consumption assays (*Figure 3F*), confirming the efficacy of the devaluation procedure in both the IPSI and the CONTRA groups (Devaluation, F(1,12) = 86.7, p<0.0001; Devaluation X Lesion, F < 1).

Collectively, these data show that disconnecting the OFC from the MD did not alter instrumental conditioning or the ability to update action-outcome contingencies. As the OFC receives a prominent innervation from another thalamic source, the submedius thalamic nucleus, which was previously suggested to interact with the OFC (*Alcaraz et al., 2015*), we used a separate cohort of rats to examine the effect of disconnecting the OFC from the Sub in the same instrumental procedure.

# Experiment 2: Disconnecting the orbitofrontal cortex from the submedius thalamic thalamus

## Instrumental training

As in Experiment 1, instrumental conditioning gradually developed over the 10 consecutive training sessions for both groups (*Figure 4A*). In this instance, instrumental performance appeared similar for both groups. The ANOVA conducted on these data support this observation as we found a significant effect of Session (F(9,153) = 74.1, p<0.0001) but not Lesion or Lesion X Session interaction (Fs <1).

## Outcome devaluation

*Figure 4b* shows the test conducted under extinction conditions, immediately after outcome devaluation. During sensory-specific outcome devaluation, all rats consumed similar amounts of food (IPSI: 5.8 ± 2.0 g, CONTRA: 7.9 ± 5.2 g, F(1,17) = 1.4, p=0.2528). As observed previously, both groups expressed a marked preference for the lever associated with the still valued outcome (F(1,17) = 29.2, p<0.0001.). Again, this preference did not vary as a function of lesion group as neither the main effect of Lesion nor that of the Lesion X Devaluation interaction were significant (Fs <1). The post-test consumption assays shown in *Figure 4C* confirmed that devaluation was effective for both groups and that rats in the IPSI and CONTRA groups consumed similar amounts of both the valued and the devalued outcomes (Devaluation, F(1,17) = 39.5, p<0.0001; Lesion, F(1,17) = 1.8, p=0.1884; Lesion X Devaluation, F(1,17) = 2.2, p=0.1527).

## Reversal

As in experiment 1, rats then received five RR20 instrumental training sessions, during which the action-outcome contingencies were reversed. During that phase, all rats maintained a high level of instrumental responding (*Figure 4D*) that did not differ between groups (Lesion, F < 1). Instrumental performance appeared fairly stable across sessions even if small daily variations likely prompted the

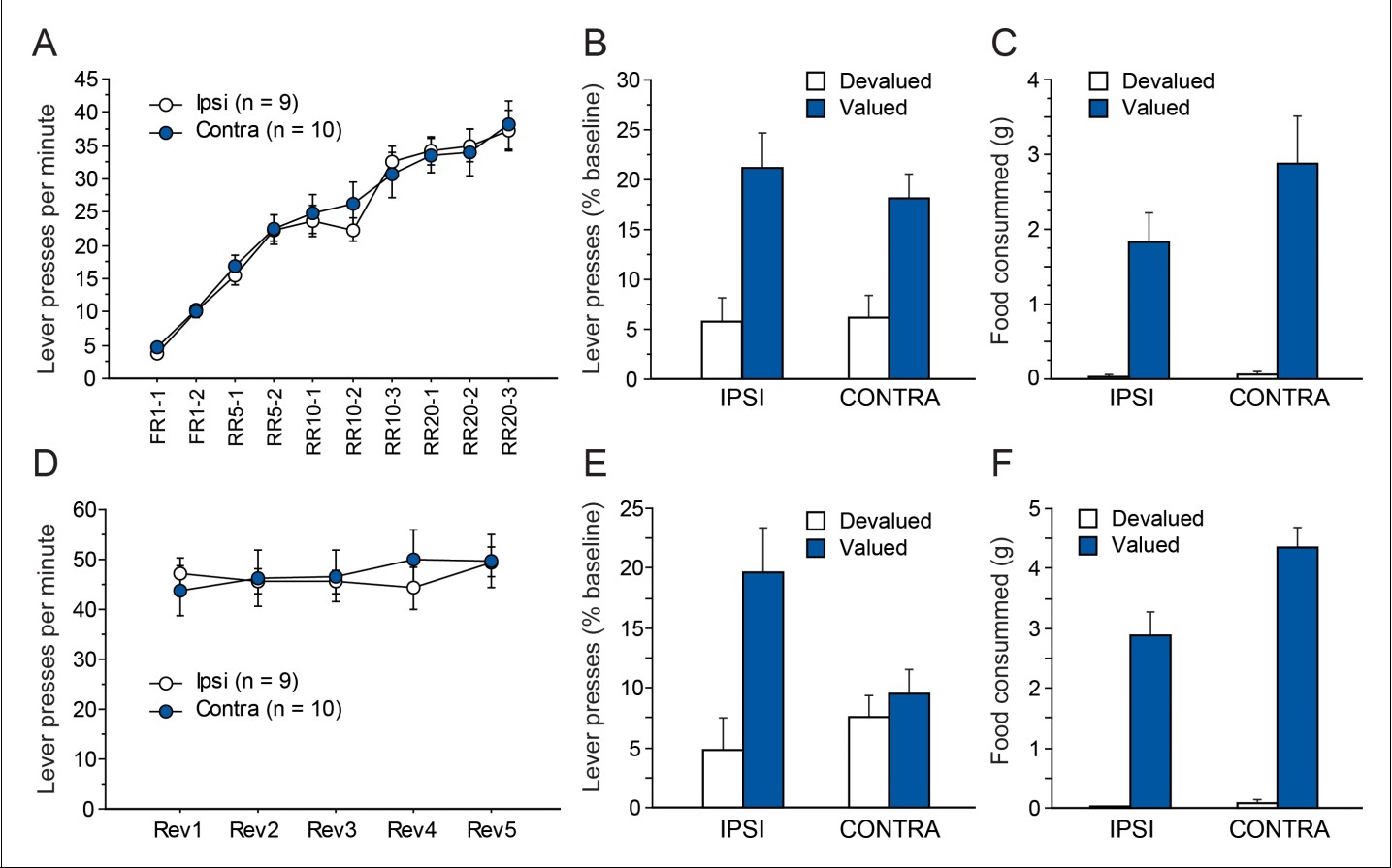

**Figure 4.** Disconnecting the OFC and the Sub. Mean rate of lever presses (± sem) during instrumental conditioning (A) or during action-outcome contingency reversal (D). Lever presses (% Baseline,+ sem) during the choice test that followed outcome devaluation immediately after initial instrumental conditioning (B) or contingency reversal (E). Consumption tests following the initial choice test (C) or the post-reversal one (F).
DOI: https://doi.org/10.7554/eLife.46187.005

main effect of Session (F(4,68) = 2.4, p=0.0609) and the Lesion X Session interaction (F(4,68) = 2.3, p=0.0643) to approach significance.

*Figure 4E* shows the results from the choice test following contingency reversal. During sensory-specific outcome devaluation, all rats consumed similar amounts of food (IPSI: 9.5 ± 3.1 g, CONTRA: 11.3 ± 3.6 g, F(1,17) = 1.3, p=0.2737). Rats from the CONTRA group appear to respond similarly for both outcomes while the IPSI group expressed a clear preference for the action earning the still valued outcome. Indeed, we found a main effect of Devaluation (F(1,17) = 10.2, p=0.0054) and a marginal effect of Lesion (F(1,17) = 4.0, p=0.0623) but also, critically, a significant Lesion X Devaluation interaction (F(1,17) = 5.6, p=0.0296). Additional analyses confirmed that Devaluation reached significance for the IPSI (F(1,8) = 11.3, p=0.0099) but not the CONTRA group (F < 1) indicating that rats from the CONTRA group were unable to display adaptive responding.

Results from the post-test consumption assay confirmed that the devaluation procedure was effective for both groups (*Figure 4F*). We found an overall effect of Devaluation (F(1,17) = 163.0, p<0.0001) and also an effect of Lesion (F(1,17) = 8.8, p=0.0085) and a significant Lesion X Devaluation interaction (F(1,17) = 4.6, p=0.0239). Nevertheless, simple effect contrasts conducted on the interaction confirmed that both groups consumed significantly more of the valued outcome than the devalued outcome (Devaluation, IPSI: F(1.8) = 53, p<0.0001), CONTRA: F(1,9) = 117.3, p<0.0001).

In summary, the behavior displayed by rats from the CONTRA group during the final test after reversal was in clear contrast with that exhibited during the prior choice test, suggesting an inability to update action-outcome contingency.

# Discussion

In a recent study, we showed that functional interactions between the mediodorsal thalamus (MD) and the medial prefrontal cortex are essential for goal-directed decision-making (*Alcaraz et al., 2018*). As the MD also connects with the orbitofrontal cortex (OFC), a region now known to play a role in goal-directed behavior (*Parkes et al., 2018*), the present study primarily aimed to extend our prior findings by focusing on MD-OFC interactions. We found that disconnecting the OFC from the MD produced no detectable impairment, even after reversal of action-outcome contingencies. This prompted us to examine whether thalamocortical interactions between the OFC and its other main thalamic afferent, the submedius thalamic nucleus (Sub), could support performance in this situation. The results clearly indicated a specific role for the OFC-Sub circuit in guiding choice after action-outcome contingency reversal. Taken as a whole, the data suggest an important role for this thalamocortical circuit in goal-directed behavior and further underscore the functional importance of a still poorly known thalamic region.

The MD is now acknowledged as a major partner of the prefrontal cortex for cognition, from rodent to primate, including humans (*Bradfield et al., 2013*; *Browning et al., 2015*; *Wolff et al., 2015*; *Parnaudeau et al., 2018*; *Pergola et al., 2018*; *Rikhye et al., 2018b*; *Wolff and Vann, 2019*). However, to date, most experimental studies have examined its functional interactions with the medial prefrontal cortex (*Bolkan et al., 2017*; *Schmitt et al., 2017*; *Alcaraz et al., 2018*; *Marton et al., 2018*; *Rikhye et al., 2018a*), thus neglecting the potential importance of its abundant connections with the OFC (*Alcaraz et al., 2016*; *Murphy and Deutch, 2018*). The present study does not support the idea that these interactions are important for goal-directed behavior, even in a setup known to engage OFC functions (*Parkes et al., 2018*). The functional relevance of the OFC-MD connections has been documented in other settings as both MD and OFC neurons were found to code for task-relevant elements in an odor-discrimination task (*Courtiol and Wilson, 2016*); however, this feature was not affected by inhibition of the corticothalamic pathway (*Courtiol et al., 2019*). In the primate, disconnecting the MD from the ventrolateral and orbital cortex impairs adaptive decision-making after selective outcome devaluation (*Browning et al., 2015*). By contrast, we observed the same effect after disconnecting the MD from the medial prefrontal cortex (*Alcaraz et al., 2018*) but not the OFC (present data). This certainly calls for further work to document the functional equivalence of prefrontal territories in rodent and primate (*Laubach et al., 2018*), especially when the functional parcellation of the OFC is now becoming clearer (*Izquierdo, 2017*; *Panayi and Killcross, 2018*). Thus, more work is needed to identify the functions that may be supported by connections between the OFC and the MD. An attractive prospect could be to rely on situations where choice is guided by stimulus-outcome associations, as previously suggested (*Ostlund and Balleine, 2008*; *Wolff et al., 2015*; *Alcaraz et al., 2016*).

Disconnecting the OFC from the Sub did not impair the ability to learn action-outcome associations and to use this information to guide choice. However, when action-outcome associations were reversed, rats with contralateral lesions of the OFC-Sub pathway were unable to integrate this new information to guide subsequent choice between the two actions. This does not necessarily imply that these animals were unable to notice these changes during the reversal phase. If that was the case they would show a devaluation effect consistent with the original contingencies, which was not observed during that test. Instead, reduced responding was observed for both actions, suggesting that these rats could not accurately segregate multiple action-outcome contingencies. It is also possible that reversal learning was incomplete. Indeed, the current study did not assess whether contralateral lesioned rats would learn the reversed contingencies with additional training but even if that was the case, it is clear that reversal learning is at least slowed down by OFC-Sub disconnection. Both hypotheses therefore point to a specific deficit in updating contingency information rather than an overall deficit in learning the relationship between actions and their outcomes. This is consistent with a prior study indicating a deficit in the ability to update stimulus-outcome contingencies in Sub lesioned rats (*Alcaraz et al., 2015*). Overall, this suggests that connections between the OFC and the Sub are critical to distinguish between learning experiences or task states (*Wilson et al., 2014*). It is also consistent with the emerging view that the cognitive thalamus critically assists the cortex to ensure that mental representations or cognitive maps are still relevant (*Wolff and Vann, 2019*).

A limitation of the present approach however lies in the lesion disconnection procedure. While targeting projection-defined cortical and thalamic neurons previously enabled us to differentiate the

functional contribution of thalamocortical and corticothalamic pathways (*Bolkan et al., 2017*; *Alcaraz et al., 2018*), it is not possible with this lesion approach to assign specific functions to OFC-to-Sub and Sub-to-OFC pathways. Moreover, while thalamocortical projections are known to be almost exclusively ipsilateral, returning corticothalamic pathways cross to the other hemisphere (*Négyessy et al., 1998*). Thus, the latter would not be compromised with the present approach. Keeping this in mind, we cannot formally exclude that targeting more efficiently both thalamocortical and corticothalamic pathways would produce different results when addressing the OFC-MD circuit. By contrast, this further clarifies the involvement of the OFC-Sub circuit, as the remaining contralateral corticothalamic pathways were not sufficient to support the updating of instrumental contingencies.

Together with prior evidence concerning the MD (*Alcaraz et al., 2018*) and the Sub (*Alcaraz et al., 2015*), the present set of data clarify the nature of the functional interactions between cortical and thalamic areas. First, distinct thalamocortical circuits appear to support instrumental performance, depending on whether updating of the associative structure of the task is required: after initial instrumental training, functional interactions between the mPFC and the MD but not the OFC and the Sub are necessary to guide choice based on current goal value but the exact opposite pattern is found after action-outcome reversal. In other words, there is striking double dissociation regarding the functional involvement of the mPFC-MD and the OFC-Sub circuits in instrumental performance. This major result fits well with novel conceptions of the functioning of thalamocortical circuits, emphasizing parallel and synergetic information processing to achieve cognitively relevant functions (*Rikhye et al., 2018b*; *Wolff and Vann, 2019*). Second, the present data are in agreement with a broader conception of OFC functions but they also highlight the contribution of a poorly understood thalamic region. The Sub indeed appears to be important to update not only stimulus-outcome but also action-outcome associations, consistent with the idea that thalamic nuclei may help to shape mental representations (*Wolff and Vann, 2019*). Finally, it would be very useful to document the functions that may be supported by OFC-MD connections. As thalamic afferents from both the Sub and the MD terminate within the same OFC loci (*Alcaraz et al., 2015*; *Alcaraz et al., 2016*), this would provide an ideal venue to investigate the functional relevance of convergence in cognition (*Man et al., 2013*).

In conclusion, the present study complements prior findings to broaden our understanding of the functioning of thalamocortical circuits. Rather than acting as isolated loops assigned to specific functions, they would be better described as supporting each other in the face of changing circumstances. An influential view posits that cortical areas may communicate directly but also indirectly through the thalamus (*Sherman, 2016*; *Sherman, 2018*; *Usrey and Sherman, 2019*). One consequence of this functional organization is that convergent and divergent thalamocortical and corticothalamic pathways may help to recruit the currently relevant circuit and/or to gate the most meaningful inputs (*Wolff and Vann, 2019*). Thus, ensuring the accuracy of mental representations over time, and over changing circumstances, likely necessitates the cooperation of multiple circuits. As noted previously, this suggests cautious use of causality concepts to interpret data in contemporary neurosciences (*Yoshihara and Yoshihara, 2018*). Thalamic research has now entered a new era and we can expect in the coming years a more systematic assessment of thalamocortical circuits that should provide important insights to better apprehend mental conditions best described as connectivity disorders such as schizophrenia (*Anticevic et al., 2014*), obsessive-compulsive behaviors (*Greenberg et al., 2010*; *Monteiro and Feng, 2016*) or addiction (*Balleine et al., 2015*).

## Materials and methods

### Animals and housing conditions

48 male Long Evans rats weighing 275 g to 300 g at surgery were obtained from Centre d'Elevage Janvier (France). Rats were initially housed in pairs and accustomed to the laboratory facility for two weeks before the beginning of the experiments. Environmental enrichment was provided by tinted polycarbonate tubing elements, in accordance with current French (Council directive 2013–118, February 1, 2013) and European (directive 2010–63, September 22, 2010, European Community) laws and policies regarding animal experiments. The facility was maintained at 21 ± 1°C with lights on from 7 a.m. to 7 p.m. The experimental protocols received approval #5012053-A from the local

Ethics Committee on December 7, 2012. After histological verification (see below), the final group sizes were: IPSI n = 7, CONTRA n = 7 for the OFC-MD disconnection and IPSI n = 9, CONTRA n = 10 for the OFC-Sub disconnection.

## Surgery

Rats were anaesthetized with 4% Isoflurane and placed in a stereotaxic frame with atraumatic ear bars (Kopf, Tujunga, CA) in a flatskull position. Anaesthesia was maintained with 1.5–2% Isoflurane and complemented by subcutaneous administration of buprenorphin (Buprecare, 0.05 mg/kg). To disconnect the OFC from its two main thalamic afferents, unilateral neurotoxic lesions were performed in contralateral hemispheres using multiple NMDA micro-injections. 20 µg/µl NMDA (Sigma-Aldrich) in artificial cerebrospinal fluid (CMA Microdialysis AB, Solna, Sweden) was pressure injected (Picospritzer, General Valve Corporation, Fairfield, NJ) into the brain through a glass micropipette (outside diameter approximately 100 µm) and polyethylene tubing. For OFC lesions, three injections sites per side (0.1 µL each) were used: AP +4.2,+3.7, and +3.2 mm from bregma, laterality ±1.2, 2.2 and 3.0 mm, DV −4.4,–4.5 and −5.2 mm from Bregma. To perform neurotoxic MD lesions, one injection site (0.15 µL) was used at the following coordinates: AP: −2.7; Lat:±0.8; DV: −5.0 (from dura). Neurotoxic Sub lesions were made using the same procedure, with one injection site per side (0.06 µL) at the following coordinates: AP: −2.7; L:±0.7; DV: −7.1 (from dura). In all cases, the pipette was left in place 3 min after injection before slow retraction. For CONTRA groups, cortical and thalamic lesions were performed on different hemispheres while lesions were made in the same hemisphere for the IPSI groups. Importantly, the extent of cortical and thalamic damage is expected to be similar in both conditions. The lateralization of the lesions was counterbalanced across all groups and conditions (for example, in the OFC-MD-CONTRA group, half the animals had the cortical lesions in the left side and half on the right side). Rats were given at least 10 days of recovery before behavioral testing.

## Behavioral experiments

### Behavioral apparatus

Animals were trained in eight identical conditioning chambers (40 cm wide x 30 cm deep x 35 cm high, Imetronic, France), each located inside a sound and light-attenuating wooden chamber (74 × 46×50 cm). Each chamber had a ventilation fan producing a background noise of 55 dB and four LEDs on the ceiling for illumination of chamber. Each chamber had two opaque panels on the right and left sides, two clear Perspex walls on the back and front sides and a stainless-steel grid floor (rod diameter: 0.5 cm; inter-rod distance: 1.5 cm). In the middle of the left wall, a magazine (6 × 4.5×4.5 cm) received either grain or sucrose pellets (45 mg, F0165 and F0023, Bio Serv, NJ, USA) from dispensers located outside the operant chamber. The magazine was equipped with infra-red cells to detect the animal's visits. Two retractable levers (4 × 1×2 cm) could be inserted on the left and right of the magazine. Activation of either the left or the right lever produced the delivery of the associated outcome, as a function of the current procedure (i.e. FR1, RR5, RR10 or RR20, see below). A computer connected to the operant chambers and equipped with POLY software and interface (Imetronic, France) controlled the equipment and recorded the data.

### Instrumental training

Rats were first habituated to the magazine dispenser through two daily sessions of magazine training for 2 days. A session consisted in the delivery of 30 food rewards, grain or sucrose pellets, distributed randomly throughout a 30 min session. The first session took place in the morning, and the second in the afternoon, with the order of rewards counterbalanced between rats and days. After magazine training, instrumental conditioning began for a total of ten daily sessions, during which rats had to learn the specific, causal association between two responses (left or right lever presses) and the two different outcomes (grain or sucrose pellets). For each daily instrumental session, each lever was presented twice until 10 min elapsed or 20 rewards were earned in an alternating fashion (e.g., lever 1 → lever 2 → lever 1 → lever 2). Thus, the session could last up to 40 min and the rats could earn a maximum of 80 rewards. The action-outcome associations and the order of their presentations were counterbalanced between rats and days. For the two first sessions, each action was reinforced (FR1). Then, for the next two sessions, a random ratio schedule of 5 was introduced

(probability of receiving an outcome given a response = 0.2). Sessions 5 to 7 were performed with a RR10 schedule (probability of receiving an outcome given a response = 0.1) and sessions 8 to 10 with a RR20 (probability of receiving an outcome given a response = 0.05). The last instrumental session with each action was used as a measure of baseline performance for the devaluation test. The entire behavioral procedure is depicted in *Figure 2*.

## Outcome devaluation test

The day after the last session of instrumental training, rats were placed in a plastic feeding cage containing free access to 15 g of one of the two outcomes for one hour of specific satiety-induced devaluation. Half of the rats in each action-outcome assignment received grain pellets, the remaining received sucrose pellets. Immediately after, rats were put in the operant cages for a 10 min extinction test. During the test, both actions were available but were unrewarded. This ensured that rats were using representations of the action-outcome contingencies and outcome value to guide their behavior. Performance was quantified relative to prior baseline levels as our previous study (*Alcaraz et al., 2018*). After one test, a second test was conducted during which the identity of the devalued outcome was now reversed so that all rats were tested after devaluation of either outcome.

## Consumption test

After the extinction test, rats were returned to the plastic feeding cages. They had free access to 10 g of each outcome for 10 min. Food consumed was then measured for each outcome to verify that the devaluation procedure was effective.

## Reversal of the instrumental contingencies

The procedure used was adapted from a previous study (*Parkes et al., 2018*). Following outcome devaluation testing, the same rats then underwent reversal training such that they were required to learn the reversed instrumental contingencies (e.g., the left lever now earned sugar pellets rather than grain pellets and the right lever now earned grain pellets rather than sugar pellets). Reversal training sessions were otherwise identical to initial instrumental training. Rats received five reversal sessions in total on an RR20 schedule of reinforcement. Outcome devaluation tests were conducted after reversal training in the same manner as that previously described. Consumption tests were also conducted after each instrumental test, as previously described.

## Histology

Animals received a lethal dose of sodium pentobarbital and were perfused transcardially with 150 ml of saline followed by 200 ml of 10% formalin. The sections throughout cortical and thalamic regions of interest were collected onto gelatin-coated slides and dried before being stained with thionine. Histological analysis of the lesions was performed under the microscope by two experimenters (VF and MW) blind to lesion conditions.

## Data analysis

The data were submitted to ANOVAs on StatView software (SAS Institute Inc). For both experiments, Lesion (IPSI/CONTRA) was the between subject factor, and Devaluation (Devalued/Valued), and Session (averaged over both actions) were repeated measures when appropriate. The alpha value for rejection of the null hypothesis was 0.05.

## Additional information

### Funding

| Funder | Grant reference number | Author |
|---|---|---|
| Agence Nationale de la Recherche | ANR-14-CE13-0029 | Mathieu Wolff |
| Brain Research Foundation | 27402 | Mathieu Wolff |

| Centre National de la Recherche Scientifique | 07803 | Shauna L Parkes<br>Etienne Coutureau<br>Mathieu Wolff |

The funders had no role in study design, data collection and interpretation, or the decision to submit the work for publication.

## Author contributions
Virginie Fresno, Data curation, Formal analysis, Investigation, Methodology; Shauna L Parkes, Etienne Coutureau, Conceptualization, Funding acquisition, Methodology, Writing—review and editing; Angélique Faugère, Investigation, Methodology; Mathieu Wolff, Conceptualization, Supervision, Funding acquisition, Investigation, Methodology, Writing—original draft, Project administration

## Author ORCIDs
Shauna L Parkes (iD) http://orcid.org/0000-0001-7725-8083
Mathieu Wolff (iD) http://orcid.org/0000-0003-3037-3038

## Ethics
Animal experimentation: Environmental enrichment was provided in accordance with current French (Council directive 2013-118, February 1, 2013) and European (directive 2010-63, September 22, 2010, European Community) laws and policies regarding animal experiments. The experimental protocols received approval #5012053-A from the local Ethics Committee on December 7, 2012.

## Decision letter and Author response
Decision letter https://doi.org/10.7554/eLife.46187.009
Author response https://doi.org/10.7554/eLife.46187.010

# Additional files

## Supplementary files
• Transparent reporting form
DOI: https://doi.org/10.7554/eLife.46187.006

## Data availability
All data generated or analysed during this study are included in the manuscript and supporting files.

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
