## [Decision Letter]

Thank you for submitting your article "A thalamocortical circuit for flexible goal-directed action" for consideration by *eLife*. Your article has been reviewed by three peer reviewers, including Geoffrey Schoenbaum as the Reviewing Editor and Reviewer #1, and the evaluation has been overseen by Timothy Behrens as the Senior Editor. The following individual involved in review of your submission has agreed to reveal their identity: Laura Corbit (Reviewer #3).

The reviewers have discussed the reviews with one another and the Reviewing Editor has drafted this decision to help you prepare a revised submission.

Summary:

This is an exceptional study that explores the role of OFC-thalamic circuits on the acquisition and use of flexible associative representations, using an instrumental reinforcer devaluation task. The authors show convincingly that a particular circuit plays a critical role in how new information is integrated and segregated with old information. The results provide novel insight into the neural circuits controlling behavior.

Essential revisions:

The reviewers agreed unanimously that the study was excellent. Although each had suggestions and questions, they also agreed that none were essential to publication of the results. Thus while we hope the reviews are valuable and can be used to improve the paper, none of the requests must be addressed in any particular way or even at all for acceptance.

*Reviewer #1:*

In the current study, the authors tested the effects of disconnecting the rat OFC from either the MD or submedius thalamic nucleus on changes in instrumental responding after sensory-specific outcome devaluation. They report that neither circuit was necessary for instrumental learning, sensory-specific devaluation, or for the normal, specific changes in instrumental responding caused by devaluation; however the OFC-sub circuit was necessary for maintaining this specificity after reversal of the action-outcome associations. That is, while ipsilateral lesioned controls reduced responding on the lever most recently associated with the devalued outcome, rats with OFC-sub disconnection reduced responding on both levers equally.

Overall I thought this was an exceptional study. The experiments were well designed, the results were analyzed and presented very clearly, and the findings were compelling, providing important new information about the neural networks mediating our ability to properly control behavior. I particularly like the use of the lesion approach, as I think it provides a ground truth that is less subject to uncertainties regarding expression, effects of rebound, and other issues. True there may be compensation and the effects may not reflect direct projections, but knowing the precise wiring is overemphasized in my opinion, and some of the feared "compensation" reflects the recovery of the downstream areas from the sudden loss of carrier signals from the lesioned areas. I think allowing this recovery to occur lessens the risk that the positive results may be "acute off-target effects" to coin a phrase from a paper on the subject. That makes the findings here reliable I think, which I like.

Indeed, I only have one concern, which is in how the authors characterize the effects as reflecting impaired updating of the action-outcome associations. I was not completely clear what was meant by this? But a simple interpretation is that without updating, the rats just maintained the original action-outcome mapping. Yet if this were the case, then they would have shown a devaluation effect, just in the wrong direction, and this is not really what they did. Instead the contra-lesioned rats seemed to reduce responding on both levers. This is as if they mixed up the two learning episodes or failed to update properly perhaps. Unless the authors disagree with me, I think they should clarify what they mean by updating in the discussion. And I think they should highlight this aspect of the results much more. Otherwise they are "dumbing down" the elegance of their behavior and the uniqueness of the finding. It will be lumped it in with the now-vast literature on goal-directed behavior and devaluation, which study only the initial learning. I think in some regards what is being studied here is quite different. Indeed, you might imagine the function that is lost here would be orthogonal to the type of information being represented? In any event, I think the authors are doing their results a disservice by not making this clearer.

*Reviewer #2:*

This study by Fresno et al. applies a circuit disconnection to investigate whether orbitofrontal cortex (OFC) connections to the thalamus are required for goal-directed action selection, assayed using the reward devaluation paradigm. Disconnecting the OFC and mediodorsal thalamus (MD) with contralateral neurotoxic lesions had no effect on task performance, either before or after action-outcome contingency reversal. Disconnecting the OFC from the submedius nucleus (Sub) spared rats' ability to select actions based on expected reward value after initial action-outcome learning but disrupted their ability to do so following contingency reversal, which is consistent with the behavioral deficits produced by bilateral OFC inactivation, as previously shown by this group.

This is a well conceived and executed study. The scope of this study is relatively narrow but addresses an important question. The design is generally solid, though the specific disconnection approach used has some limitations. The results are clear and support the authors' conclusions. The manuscript is well-written and does a good job of acknowledging limitations of the study. My only reservations are with the overall significance and scope of the study.

The study provides compelling evidence that the OFC and Sub interact to support reward devaluation after contingency reversal, and I tend to agree this is an important aspect of behavioral control. But the authors could do more to justify the significance of the reversal component of the task and elaborate on why they believe it picks up something important that the standard reward devaluation task does not. This should also be addressed when discussing the findings, perhaps in terms of clinical significance. As it stands, the authors seem to take the significance of this task for granted and refer to it as "flexible" goal-directed action. This makes some sense but is also somewhat confusing since goal-directed actions are inherently flexible. It also fails to make clear whether this is a fundamental feature of goal-directed behavior or why it is specifically interesting.

As noted, there are some limitations to the circuit disconnection manipulation, which the authors do a good job of acknowledging. Because OFC projections to thalamus are bilateral, asymmetrical lesions allow for compensation via crossed corticothalamic circuitry. Null effects in the OFC-MD disconnection group and in pre-reversal devaluation performance for the OFC-Sub group are therefore not conclusive. Moreover, this approach does not allow the authors to tease apart the specific contributions of corticothalamic and thalamocortical projections, which the authors' own recent work sets up as an important question. The use of permanent pre-training lesions also limits the scope of the study, since the current findings do not address whether this circuitry specifically contributes to acquisition (contingency learning or remapping) or action selection processes. As the authors note, more advanced techniques like opto- and chemogenetics can overcome these limitations. Importantly, the authors main finding that crossed OFC and Sub lesions disrupt post-reversal reward devaluation performance still stands, and does indicate that this effect is particularly dependent on ipsilateral OFC-Sub circuit. Therefore, the comments above speak to the significance and scope of the current study but do not undermine the authors' main conclusions.

*Reviewer #3:*

This manuscript by Fresno et al. examines the functional role of thalamic inputs from the mediodorsal thalamus and submedius nucleus to the orbitofrontal cortex (OFC) in the acquisition and reversal of response-outcome (R-O) learning. Using a crossed lesion approach, they find that disconnection of MD and OFC leaves response-outcome learning and its reversal intact. However, functional disconnection of the submedius nucleus and OFC leaves initial learning intact; animals show a selective devaluation effect, but the same animals no longer show a selective devaluation effect once the original R-O contingencies have been reversed.

While the lesion approach is not as elegant for targeting specific pathways as recent work by this group with DREADDs, the apparently normal performance of all groups in the initial devaluation tests and specific deficits only after reversal points to a selective deficit and many typical concerns about lesions can be dismissed. The studies are well run and the manuscript is clearly written and so my overall view is that this tells us something new and important about the role of the submedius nucleus in response-outcome learning and contributes to understanding of complex thalamocortical circuits.

I have several comments for the authors to consider.

While the consumption tests provide compelling evidence that the devaluation treatment itself was effective, the consumption in the induction of devaluation (1 hour feeding period) is not reported. It would be good to know that consumption was equivalent in the two groups and whether there was a criterion for exclusion applied. It's difficult to imagine a scenario where consumption was equivalent in test 1 but differed in test 2 but based on the main effect of lesion (panel 4F) it would nonetheless be reassuring to know consumption was equivalent in the two groups.

I thought the authors could have gone deeper into what the deficit in the submedius-OFC disconnection group means. If these animals couldn't update at all, you might expect to see preferential responding for the devalued outcome. The indiscriminate responding suggests perhaps they are beginning to update, although slower than controls. I wondered whether we might see something interesting in individual data; e.g., this might address whether all rats respond similarly for the two outcomes or whether some have reversed effectively while others have not (but the means blur this). There's the possibility that individual differences could correspond to the extent or other variation in the lesions. Future studies (not needed here) could explore whether they're really unable to update at all vs do so but more slowly (e.g. would they reverse with more training?). Generally, the authors interpret their findings as a failure to update action-outcome contingencies which I think their data support. They might rethink the statement (opening sentence of Discussion paragraph three) that the lesions prevented goal-directed actions, since responding could be set on a goal, but based on an outdated R-O.

---

## [Author Response]

Essential revisions:The reviewers agreed unanimously that the study was excellent. Although each had suggestions and questions, they also agreed that none were essential to publication of the results. Thus while we hope the reviews are valuable and can be used to improve the paper, none of the requests must be addressed in any particular way or even at all for acceptance.

First, we’d like to thank all referees for their positive and constructive comments on the initial draft of this manuscript. We found all their suggestions very helpful and they all pointed towards the same general issue with the manuscript. All three reviewers indeed agreed that the major finding of the study required further discussion. As such, we have included the following paragraph in the Discussion to clarify our interpretation of the results, the title was also modified to reflect this clarification.

“Disconnecting the OFC from the Sub did not impair the ability to learn action-outcome associations and to use this information to guide choice. However, when action-outcome associations were reversed, rats with contralateral lesions of the OFC-Sub pathway were unable to integrate this new information to guide subsequent choice between the two actions. This does not necessarily imply that these animals were unable to notice these changes during the reversal phase. If that was the case they would show a devaluation effect consistent with the original contingencies, which was not observed during that test. Instead, reduced responding was observed for both actions, suggesting that these rats could not accurately segregate multiple action-outcome contingencies. It is also possible that reversal learning was incomplete. Indeed, the current study did not assess whether contralateral lesioned rats would learn the reversed contingencies with additional training but even if that was the case, it is clear that reversal learning is at least slowed down by OFC-Sub disconnection. Both hypotheses therefore point to a specific deficit in updating contingency information rather than an overall deficit in learning the relationship between actions and their outcomes. This is consistent with a prior study indicating a deficit in the ability to update stimulus-outcome contingencies in Sub lesioned rats (Alcaraz et al.et al., 2015). Overall, this suggests that connections between the OFC and the Sub are critical to distinguish between learning experiences or task states (Wilson et al.et al., 2014). It is also consistent with the emerging view that the cognitive thalamus critically assists the cortex to ensure that mental representations or cognitive maps are still relevant (Wolff and Vann, 2019).”

Reviewer #3:[…] While the consumption tests provide compelling evidence that the devaluation treatment itself was effective, the consumption in the induction of devaluation (1 hour feeding period) is not reported. It would be good to know that consumption was equivalent in the two groups and whether there was a criterion for exclusion applied. It's difficult to imagine a scenario where consumption was equivalent in test 1 but differed in test 2 but based on the main effect of lesion (panel 4F) it would nonetheless be reassuring to know consumption was equivalent in the two groups.

The consumption during devaluation and the corresponding statistics have been added in subsections of Experiments 1 and 2 “Outcome Devaluation” and “Reversal”. The only difference found was for Experiment 1 (OFC-MD disconnection) with initial contingencies, as rats from the IPSI group consumed more food than rats from the CONTRA group. But they behave the same during the subsequent choice test. On all other instances, we did not find any difference between IPSI and CONTRA (both experiments).

Future studies (not needed here) could explore whether they're really unable to update at all vs do so but more slowly (e.g. would they reverse with more training?).

The following has been added to the Discussion section: “It is also possible that reversal learning was incomplete. Indeed, the current study did not assess whether contralateral lesioned rats would learn the reversed contingencies with additional training but even if that was the case, it is clear that reversal learning is at least slowed down by OFC-Sub disconnection.”

Generally, the authors interpret their findings as a failure to update action-outcome contingencies which I think their data support. They might rethink the statement (opening sentence of Discussion paragraph three) that the lesions prevented goal-directed actions, since responding could be set on a goal, but based on an outdated R-O.

This statement has been changed to: “Disconnecting the OFC from the Sub did not impair the ability to learn action-outcome associations and to use this information to guide choice. However, when action-outcome associations were reversed, rats with contralateral lesions of the OFC-Sub pathway were unable to integrate this new information to guide subsequent choice between the two actions.”